# Predicting Long-Term Human Behaviors in Discrete Representations via Physics-Guided Diffusion

Zhitian Zhang
Simon Fraser University
zhitianz@sfu.ca

Anjian Li
Princeton University
anjianl@princeton.edu

Angelica Lim
Simon Fraser University
aneglic@sfu.ca

Mo Chen
Simon Fraser University
mochen@sfu.ca

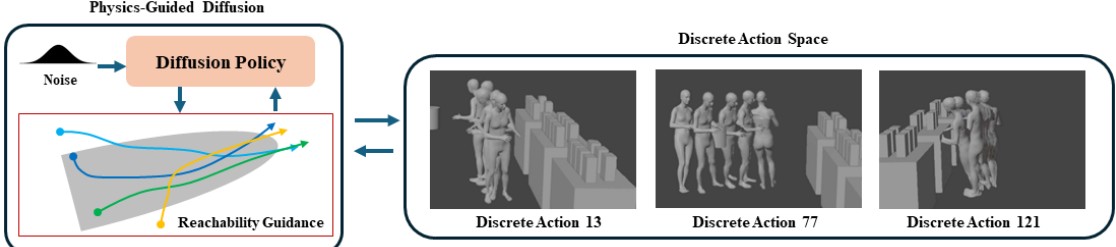

Figure 1. We propose a reachability-guided diffusion model (Left) for generating long-term human behaviors. Our model works in the discrete action space (Right). We visualize a few examples of learned discrete actions from continuous trajectory space using VQ-VAEs. Visualization is done using the SSN 3D virtual human platform [3].

## Abstract

*Long-term human trajectory prediction is a challenging yet critical task in robotics and autonomous systems. Prior work that studied how to predict accurate short-term human trajectories with only unimodal features often failed in long-term prediction. Reinforcement learning provides a good solution for learning human long-term behaviors but can suffer from challenges in data efficiency and optimization. In this work, we propose a long-term human trajectory forecasting framework that leverages a guided diffusion model to generate diverse long-term human behaviors in a high-level latent action space, obtained via a hierarchical action quantization scheme using a VQ-VAE to discretize continuous trajectories and the available context. The latent actions are predicted by our guided diffusion model, which uses physics-inspired guidance at test time to constrain generated multimodal action distributions. Specifically, we use reachability analysis during the reverse denoising process to guide the diffusion steps toward physically feasible latent actions. We evaluate our framework on two publicly available human trajectory forecasting datasets: SFU-Store-Nav and JRDB, and extensive experimental results show that our framework achieves superior performance in long-term human trajectory forecasting.*

## 1. Introduction

The ability to anticipate future behavior is a remarkable yet common skill for humans. We can navigate naturally through complex environments and perform long-term planning. Having an agent that is capable of predicting realistic and plausible human behaviors is essential to human-robot interaction and autonomous systems [32, 44]. Such prediction ability could not only benefit the decision-making of an autonomous system, but also generate high-quality data that mimics human behaviors. However, creating such an agent has been a challenging problem in the domain of machine learning and robotics.

Reinforcement learning (RL) has been widely studied in behavior learning, but can also be problematic due to reward constraints [37, 66], data limitations [61], high costs of sample complexity [13, 33] and relatively short-horizon behaviors [16, 23]. Imitation learning (IL) is a type of offline RL [25], where an agent is trained to mimic the actions

from an offline human demonstration dataset. IL has shown promising performance in autonomous driving [7], robotics [14], and game-playing [38] on imitating human behaviors. However, IL faces challenges in producing diverse distributions and long-term predictions with only mean squared error (MSE) as the optimization objective. In this work, we seek to model a diverse yet realistic human agent's behavior distribution for long-term trajectory prediction. While an agent often faces an uncountable set of states in the continuous trajectory space, our key insight is that by tokenizing the continuous trajectory space, generative models can better capture the multimodality of the distribution while avoiding the curse of dimensionality. We design a hierarchical action quantization (HAQ) scheme based on vector quantized variational autoencoder (VQ-VAE) [54] to learn the mapping between the continuous trajectory space and discrete latent action space. We then apply imitation learning in the discrete latent action space.

There are several recent advances in generating human behaviors with discrete representations [11, 28, 29, 45, 64], mostly based on autoregressive models using transformers [55]. These autoregressive models have two major drawbacks in the imitation learning setting. First, the learned policy with a distribution shift at test time could cause the agent to drift away from optimal states [41]. Second, the autoregressive model only predicts one token at a time, making it slow in generating long sequences [8]. Recently, diffusion models have been studied to imitate human behaviors [39, 59] or robot actions [9], showing a strong capability of modeling complex action distributions and a stable training process.

In this paper, we use a denoising diffusion probabilistic model (DDPM) [19] with the *analog bit* approach [8] to achieve accurate long-term human trajectory prediction. Our proposed diffusion model takes states (motion history, environmental information) as the conditional input and samples future discrete latent actions, which are learned from our hierarchical VQ-VAEs. We hypothesize that diffusion models as expressive generative models, with an efficient discrete representation of continuous trajectory space, can generate plausible human behaviors in long-horizon. To the best of our knowledge, this is the first work to generate discrete latent human actions with diffusion models. Furthermore, we propose *reachability guidance* to improve the physical feasibility of sampled trajectories. Our reachability guidance does not require the training of an extra classifier in the pipeline which could be cumbersome [18]. We demonstrate that with the proposed physics-guided diffusion policy, our approach significantly improves the performance in long-term human trajectory prediction tasks.

To summarize, our contribution is threefold. **(1)** We propose a simple yet efficient approach using a hierarchical VQ-VAE to quantize a continuous human trajectory space

to a discrete set of actions. Our discrete representations capture the multimodality of behaviors and enable the use of IL methods to generate long-term realistic human behaviors. **(2)** We present a novel paradigm where we model future human behavior distributions in discrete action space using a denoising diffusion probabilistic model (DDPM)-based behavior generator. **(3)** We introduce reachability guidance: an intuitive physics-inspired guidance that incorporates the laws of physics and safety into the denoising process, allowing our model to generate physically feasible human behaviors (Figure 1).

## 2. Related Work

### 2.1. Off Policy Learning

Off-policy learning has proven to be effective in trajectory generation. Offline imitation learning (IL) [4, 10, 34, 66] has emerged as a popular method for learning human actions from pre-collected datasets. When the reward information is available, offline reinforcement learning (RL) [15, 22, 25, 60] has also been explored for learning human actions. However, IL, as a form of supervised learning, often assumes a unimodal action distribution [24] that rarely holds in real-world datasets. To address this issue, positive unlabeled learning [58] is used to focus learning on the expert part of the dataset, thereby reducing the modality in the learning process, and behaviour transformers [45] use transformers with k-means clustering to model multiple modes of human behaviors. In addition, most existing IL methods suffer from covariant shift when applied to online trajectory prediction, limiting the effectiveness in long-term prediction [5, 35, 43, 44]. Online RL has been integrated with IL to address distribution shift through a closed-loop learning fashion [27] but it requires extensive online training time.

### 2.2. Discretizing continuous action space

Although real-world human actions occur in a continuous space, directly learning continuous actions from a limited pre-collected dataset often results in unsatisfactory performance [29]. Therefore, action space discretization has been investigated while avoiding the exponential growth of the state dimension. One common approach treats each action dimension as independent [6, 50, 51, 57], while an alternative approach is sequential discretization through learned causal dependence [31, 52, 53]. Leveraging human demonstration for action quantization has also been proven effective in generating a discrete set of reasonable actions [11]. Recently, vector quantized variational autoencoders (VQ-VAE) have been adopted for off-policy learning [51], resulting in a small discrete action space while improving the learning performance. Furthermore, action discretization plays an important role in hierarchical learning methods to help identify sub-goals and facilitate action generation over

long horizons [21].

## 2.3. Diffusion Models

Diffusion models [46, 48] are a class of generative models that are designed to sample from complex distributions through a reverse stochastic differential equation (SDE) process. Later on, the development of denoising diffusion probabilistic models (DDPM) [19] and denoising diffusion implicit models (DDIM) [47] showed remarkable efficacy in image generation tasks. To enhance the conditional input in the generation process, classifier guidance [12] and classifier-free guidance [18] are introduced. Moreover, diffusion models also demonstrate their exceptional ability to sample behaviors or trajectories from multimodal distributions in the field of robotics [1, 9, 20, 39, 49]. With the attention mechanism first proposed in DDPM [19], diffusion models also show superior performance in modeling sequential correlation of the data.

## 3. Background

### 3.1. Problem Formulation

Our objective is to learn a robust policy that could generate long-term plausible human behaviors given the current and previous states in the last $H$ time steps. In this paper, we are particularly interested in human navigational behaviors. We aim to generate a long-term future trajectory for the target agent over the next $T$ time steps. Formally, this means learning the future trajectory conditional distribution $p(\mathbb{X}|X, C)$, where $\mathbb{X} = \{x^{t+1}, y^{t+1}, \cdots, x^{t+T}, y^{t+T}\}$ represents the future trajectory, $X = \{x^{t-H}, y^{t-H}, \cdots, x^t, y^t\}$ the past trajectory including the current state, and $C = \{c^{t-H}, \cdots, c^t\}$ any other possible contextual observation features such as information related to the scene or the other nearby agents. We define the state of the agent at a given time $t$ to be $s^t = (x^t, y^t, c^t)$, with $(x, y)$ representing position. A detailed time horizon is shown in Figure 2.

As mentioned earlier, we incorporate physics priors during the learning process to encourage feasible human behavior generation. To facilitate the use of the proposed control-theoretic methods, we use a dynamically extended Dubins Car to approximately model the dynamics of a human:

$$\dot{x} = v \cos\theta, \;\; \dot{y} = v \sin\theta$$
$$\dot{v} = u_1, \;\; \dot{\theta} = u_2 \tag{1}$$

where $v$ is the speed of movement and $\theta$ is the orientation of the human. The state of the above system $(x, y, v, \theta)$ represents the augmented human state (ignoring the context $c$) used only for the purposes of incorporating physics priors. We define $\hat{u} = (u_1, u_2)$ as the control variables of our

system, where $u_1$ is the acceleration, $u_2$ is the angular velocity.[1]

## 3.2. Imitation Learning (IL)

Imitation learning constructs an optimal policy by mimicking a set of expert demonstrations, without knowing any reward function. Assume the agents have access to a dataset of expert demonstrations $\mathcal{D} = \{s^0, \hat{u}^0, \cdots, s^T, \hat{u}^T\}$ produced by the expert policy $\pi_\beta$, the goal is to learn a policy $\pi$ that imitates $\pi_\beta$. The simplest approach is through behavioral cloning (BC) where the policy is trained by maximizing the log-likelihood of actions, $\mathbb{E}_{s,\hat{u} \sim \mathcal{D}}[\log \pi(\hat{u}|s)]$.

## 3.3. Vector quantized variational autoencoder

In this work, we use the VQ-VAE [54] to tokenize continuous human trajectories. It comprises an encoder that maps observations onto a sequence of discrete latent variables and a decoder that reconstructs the observations from the discrete latent variables. The encoder $E$ outputs a continuous embedding $E(\mathbf{s})$ from the input space $\mathbf{s} = \{s^{t-T_{vq}}, \cdots, s^t\}$ and discretization is done by finding the index of the nearest prototype vector in the codebook $\mathbf{e}_j$ to the encoder embedding $E(\mathbf{s})$ based on the distance, where $j \in \{1, \cdots, J\}$. The process can be described as:

$$\text{Quantize}(E(\mathbf{s})) = \mathbf{e}_i \;\; \text{where} \;\; i = \arg\min_j \|E(\mathbf{s}) - \mathbf{e}_j\| \tag{2}$$

Let $\mathbf{z}$ denote the final quantized latent code for input $\mathbf{s}$, i.e., $\mathbf{z} = \mathbf{e}_i$; the decoder $D$ will reconstruct the original input $\mathbf{s}$ based on latent code $\mathbf{z}$.[2] In addition to the reconstruction objective, a codebook loss and a commitment loss are added to move codebook vectors closer to the encoder embeddings. The overall optimization objective of VQ-VAE can be described as:

$$\mathcal{L}_{vq} = \|\mathbf{s} - D(\mathbf{z})\|_2^2 + \|sg[E(\mathbf{s})] - \mathbf{z}\|_2^2 + \beta\|sg[\mathbf{z}] - E(\mathbf{s})\|_2^2 \tag{3}$$

where $sg$ stands for the stop gradient operator and $\beta$ is a hyper-parameter for the commitment loss. The parameters of the encoder and decoder are both optimized by this objective.

## 4. Method

Our goal is to generate diverse and feasible long-term human behaviors. The overall framework consists of three parts: Hierarchical Action Quantization, Action Diffusion Policy, and Reachability Guidance, which are illustrated in Figure 3. We consider a goal-oriented navigational problem

---

[1]Note that since Equation 1 is differentially flat, all other variables can be easily computed given $(x^t, y^t)$.
[2]In practice, a collection of vectors are quantized and decoded in parallel.

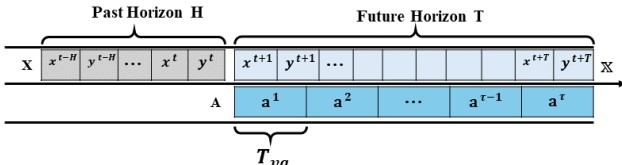

Figure 2. Illustration of past horizon $H$ and future horizon $T$ in continuous trajectory space (Top). A discrete action $\mathbf{a}^\tau$ tokenizes all states in a period of $T_{vq}$ in continuous trajectory space (Bottom).

where the human agent's objective is to navigate in the environment to reach a terminal state. In our proposed framework, our *Action Diffusion Policy* models a multimodal and high-level future action distribution $p(A|S)$ conditioned on low-level past observations $S = \{X, C\}$, where $A$ represents the predicted discrete latent action sequence in the future. Our *Hierarchical Action Quantization* maps each discrete latent action $A$ back to the continuous trajectory $\mathbb{X}$. To incorporate physics information into the reverse diffusion process, we introduce the *Reachability Guidance*. We now explain the details of each component.

## 4.1. Hierarchical Action Quantization

Naturally, human actions can be represented with discrete representations. Previous works in human trajectory prediction [30, 44, 62] mostly represent human trajectories in continuous space. However, we aim to learn a context-aware discrete representation of human high-level actions by training a VQ-VAE model.

In contrast to the vanilla VQ-VAE, in this work we propose to formulate a hierarchical action quantization (HAQ) structure to produce the discrete latent code, inspired by [40]. The motivation behind this is that we want to capture the multimodality of human actions by modeling the context information $c^t$ separately from $\{x^t, y^t\}$. In order to achieve this, we use a two-level hierarchical structure. The hierarchy contains a *top* level encoder $E_{top}$ that learns the context information $C_{vq} = \{c^{t-T_{vq}}, \cdots, c^t\}$, i.e. body pose; a *bottom* level encoder $E_{top}$ that learns the trajectory information $X_{vq} = \{x^{t-T_{vq}}, y^{t-T_{vq}}, \cdots, x^t, y^t\}$, and a HAQ decoder $D$ that reconstructs the original input $X_{vq}$. The HAQ encoding process of the network can be written as:

$$
\begin{aligned}
\mathbf{h}_{\text{top}} &= E_{\text{top}}(C_{vq}), \quad \mathbf{a}_{\text{top}} = \text{Quantize}(\mathbf{h}_{\text{top}}) \\
\mathbf{h}_{\text{bot}} &= E_{\text{bot}}(X_{vq}, \mathbf{h}_{\text{top}}), \quad \mathbf{a}_{\text{bot}} = \text{Quantize}(\mathbf{h}_{\text{bot}})
\end{aligned}
\tag{4}
$$

where $\mathbf{h}$ is the continuous latent variables obtained from the encoders, Quantize process is defined as in Equation 2, $\mathbf{a}_{bot}$ is our final discrete action representation, and later we refer it as $\mathbf{a}$ for brevity. The overall network is optimized by the objective defined in Equation 3 for both *top* and *bot-*

*tom* codebooks, with the reconstruction objective only applying to input $X_{vq}$. After training our VQ-VAE, we obtain a semantic-rich codebook that is conditioned on contextual information and we can represent a sequence of low-level continuous states with a discrete high-level action (see Figure 1(Right)).

## 4.2. Action Diffusion Policy

In this section, we introduce our diffusion policy under the imitation learning formulation. With a learned HAQ encoder, we can represent future continuous trajectory space $\mathbb{X}$ in discrete space: $A = \{\mathbf{a}^1, \mathbf{a}^2, \cdots, \mathbf{a}^\tau\}$, where $\tau = T/T_{vq}$ (See Figure 2). Given previous continuous states $S$, we can represent a behavior cloning policy as $\pi(A|S) = p(A|S)$ (subsection 3.2). We apply a conditional diffusion model to the discrete latent action space, to model $p(A|S)$. The behavior cloning policy $\pi(A|S)$ can be then optimized by the diffusion objective, which aims to sample $A$ from the same distribution as $D$ [59].

In this work, we adopt denoising diffusion probabilistic models (DDPMs) [19] as our behavior cloning policy. And to allow our diffusion policy to generate discrete actions, we use the *analog bits* approach from the bit diffusion model [8]. We start with a short introduction to diffusion models. Starting from a noisy discrete action sequence $A_k$, where $A_K \sim \mathcal{N}(\mathbf{0}, \mathbf{I})$, a sequence of $A_{K-1}, \cdots, A_0$ is predicted through $K$ iterations of denoising steps, each with a decreasing level of noise until the "clean" output $A_0$ is formed. During training, which is also called forward diffusion process, noisy input is generated as $A_k = \sqrt{\bar{\alpha}_k}A + \sqrt{1 - \bar{\alpha}_k}\epsilon$, for some variance schedule $\bar{\alpha}_k$, random noise $\epsilon \sim \mathcal{N}(\mathbf{0}, \mathbf{I})$. A denoising network $\mathcal{G}_\theta$ is trained to predict the noise that was added to the input, conditioned on some past states $S$, by minimizing the following objective:

$$
\mathcal{L}_{\text{DDPM},\theta} := \mathbb{E}_{S,A,k,\epsilon}\left[\|\mathcal{G}_\theta(S, A_k, k) - \epsilon\|_2^2\right]
\tag{5}
$$

During the reverse diffusion process, which often refers to the sampling time, with the variance schedule parameters $\alpha$ and $\sigma$, the "cleaner" input is generated as follows:

$$
A_{k-1} = \frac{1}{\sqrt{\alpha_k}}\left(A_k - \frac{1-\alpha_k}{\sqrt{1-\bar{\alpha}_k}}\mathcal{G}_\theta(S, A, k)\right) + \sigma_k\epsilon
\tag{6}
$$

**Discrete action sequence prediction.** While DDPMs are often used in continuous space for image generation, we adopt the *analog bits* [8] method to generate discrete action sequences with the same continuous diffusion models. The *analog bits* approach is twofold: First, during training, discrete data are represented by bits and then cast into real numbers, which can be directly modeled by the DDPMs. We denote this process as *int2bit*, where in our case a discrete action $\mathbf{a}^\tau$ from a codebook of size $J$ (subsection 3.3) can be represented using $n = \lceil \log_2 J \rceil$ bits, as $\{0, 1\}^n$. Then

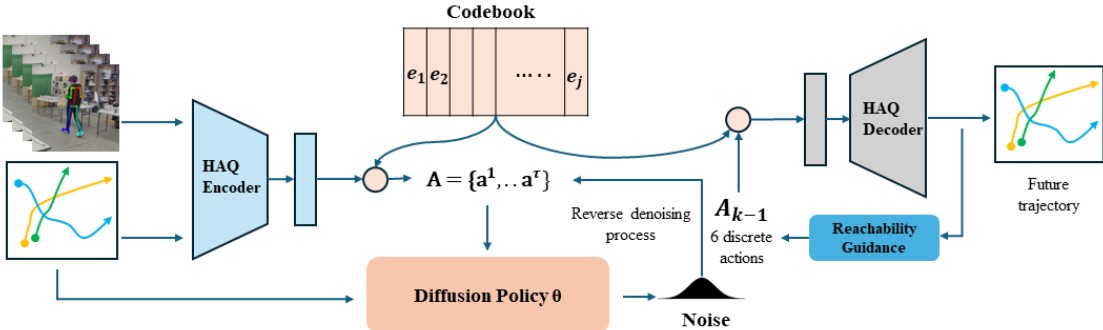

**Codebook**

$e_1$ $e_2$ ..... $e_j$

$A = \{a^1, .. a^\tau\}$

Reverse denoising process

$A_{k-1}$ 6 discrete actions

**HAQ Encoder**

**HAQ Decoder**

**Reachability Guidance**

**Diffusion Policy θ**

**Noise**

Future trajectory

Figure 3. **Overview of our framework.** Hierarchical Action Quantization (HAQ) encoder learns a discrete representation of human behaviors. Our diffusion policy generates 6 discrete future actions conditioned on past observations. During each reverse denoising process, reachability guidance is used to enforce some physical constraints. The final output is a long-term future human trajectory reconstructed from discrete future actions using the HAQ decoder.

during sampling, we draw samples following the same procedures in DDPMs, and apply a simple thresholding before decoding back into discrete data. We denote this process as *bit2int*. The whole process can be described with the following:

$$A_0 = int2bit(\mathbf{A}), \text{ forward diffusion} \quad (7)$$

$$\mathbf{A} = bit2int(A_{k-1}), \text{ reverse diffusion} \quad (8)$$

### 4.3. Reachability Guidance

Given that a diffusion policy does not inherently have knowledge on fundamental physics laws, it is highly possible to generate human behaviors that are infeasible and clearly disobey the physics world. In this work, we apply a physical constraint during the diffusion process, which we call *reachability guidance*. The motivation is simple: every step of our diffusion process produces an intermediate sequence of discrete actions $A_{k-1}$, and by applying the reachability guidance, we could guide the diffusion process towards generating samples that are physically feasible. The complete procedure is shown below:

**Reachability.** Hamilton-Jacobi (HJ) reachability analysis is a formal method that can verify the performance and safety of a dynamic system [2]. Given the assumed dynamics of the human in Equation 1, we can compute a Backward Reachable Set (BRS) based on the discrete action $\mathbf{a}^\tau$. The BRS represents the set of states such that the trajectories that start from this set can reach the given discrete action $\mathbf{a}^\tau$ (See Figure 1(Left)). To calculate the BRS, we first decode the discrete action $\mathbf{a}^\tau$ using HAQ decoder $D$ to obtain the states $\{s^{t-T_{vq}}, \cdots, s^t\}$ where $(x^t, y^t, \theta^t, v^t)$ can be derived from $s^t$, then the process can be written as:

$$\mathcal{S} = \mathbf{BRS}(D(\mathbf{a}^\tau)) \quad (9)$$

where $\mathcal{S}$ represents the possible positions that the agent could be located in order to feasibly reach the starting states

---

**Algorithm 1** Reachability guided diffusion sampling

**Input:** Past states $S$, denoising network $\mathcal{G}_\theta$, denoising timestep $k$.
**Output:** Denoised action $A_0$.
1: $A_K \sim \mathcal{N}(\mathbf{0}, \mathbf{I})$
2: **for** $k$ from $K$ to 0 **do**
3: $\quad \mu, \Sigma \leftarrow \mathcal{G}(A_k, k, S)$
4: $\quad A'_k \sim \mathcal{N}(\mu, \Sigma)$ ▷ Sample $A'_k$ without guidance
5: $\quad$ Compute $p_\gamma(r|A'_k)$ ▷ Reachability guidance
6: $\quad \tilde{\mu} \leftarrow \mu + s\Sigma\nabla_{A'_k} \log p_\gamma(r|A'_k)$ ▷ Compute the new mean
7: $\quad A_{k-1} \sim \mathcal{N}(\tilde{\mu}, \Sigma)$ ▷ Sample $A_{k-1}$ with guidance
8: **end for**
9: **return** $A_0$

---

in $\mathbf{a}^\tau$, and **BRS** represents the mathematical calculation of the backward reachable set. For the sake of brevity, we will defer to [2] for readers interested in the details of reachability. Intuitively, the BRS should cover all the states from the previous discrete action: $D(\mathbf{a}^{\tau-1}) \subseteq \mathcal{S}$ Thus, to determine whether an action is physically feasible, we can formulate a classification problem where:

$$p(\mathbf{a}^{\tau-1}) = \begin{cases} 1, & D(\mathbf{a}^{\tau-1}) \subseteq \mathcal{S} \\ 0, & \text{otherwise} \end{cases} \quad (10)$$

Then the probability that a sequence of actions $A = \{\mathbf{a}^1, \mathbf{a}^2, \cdots, \mathbf{a}^\tau\}$ is physically feasible can be easily calculated as:

$$p_\gamma(r|A) = \frac{1}{\tau - 1} \sum_{\tau'=1}^{\tau-1} p(\mathbf{a}^{\tau'}) \quad (11)$$

More generally, we can write $p_\gamma$[3] in term of any intermediate latent $A_k$, $p_\gamma(r|A_k)$. For brevity, we leave out the

---

[3]$\gamma$ does not represent network parameters here.

quantize operation as defined in Equation 7 in the reachability analysis process.

**Guidance.** Classifier guidance is a useful technique for improving diffusion models [12, 46, 48]. The process involves training an additional classifier with class labels on noisy inputs and using a classifier gradient to guide the diffusion sampling process. Similarly, one can also use the gradient of reachability "classification" to guide the diffusion process. We believe that this is a simple approach to force some physical constraint to the network, and only applied during the diffusion sampling time. Here we show a brief derivation on how to modify an unconditional reverse diffusion process $p_\theta (A_k|A_{k+1})$ to condition on the reachability analysis result $p_\gamma (r|A_k)$, where

$$p_{\theta,\gamma} (A_k|A_{k+1}, r) = Z p_\theta (A_k|A_{k+1}) p_\gamma (r|A_k) \quad (12)$$

And $Z$ is a normalizing constant. Recall that:

$$p_\theta (A_k|A_{k+1}) = \mathcal{N}(\mu, \Sigma) \quad (13)$$

$$\log p_\theta (A_k|A_{k+1}) = -\frac{1}{2} (A_k - \mu)^T \Sigma^{-1} (A_k - \mu) + C \quad (14)$$

We can approximate $\log p_\gamma (r|A_k)$ using a Taylor expansion around $A_k = \mu$:

$$\log p_\gamma (r|A_k) \approx \log p_\gamma (r|A_k)|_{A_k=\mu}$$
$$+ (A_k - \mu) \nabla_{A_k} \log p_\gamma (r|A_k)|_{A_k=\mu} \quad (15)$$

Letting $g = \nabla_{A_k} \log p_\gamma (r|A_k)|_{A_k=\mu}$, we can derive an approximation of desired distribution:

$$\log (p_\theta (A_k|A_{k+1}) p_\gamma (r|A_k)) \approx$$
$$- \frac{1}{2} (A_k - \mu)^T \Sigma^{-1} (A_k - \mu)$$
$$+ (A_k - \mu) g \quad (16)$$

And finally:

$$\log p_{\theta,\gamma} (A_k|A_{k+1}, r) = \log p(z), z \sim \mathcal{N}(\mu + \Sigma g, \Sigma) \quad (17)$$

This derivation suggests that the conditional distribution can be approximated by shifting the mean of unconditional distribution by $\Sigma g$. Following [12], a scale factor $s$ is also added to the gradient calculation.

# 5. Experiments

We evaluate our methods on human trajectory forecasting on two publicly available datasets: SFU-Store-Nav (SSN) [65] and JRDB [56] datasets. Both datasets consist of real human trajectories with associated visual information. We sample both datasets at 3Hz and split the dataset into training, validation, and testing sets with proportions of 80%,

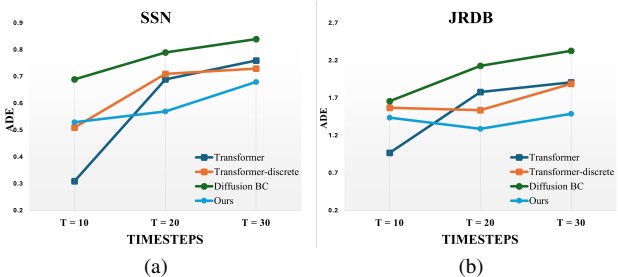

(a)             (b)

Figure 4. Comparison between different forecasting timesteps (T = 10, 20 ,30) on SSN and JRDB datasets in terms of ADE. Our model is better at generating long-term future trajectories, while still maintaining comparable performance in short-term prediction.

5%, and 15% respectively. Following prior works in human trajectory forecasting [17, 44, 67], we use two error metrics to evaluate the generated human trajectories:

- Average Displacement Error (ADE): Mean $l_2$ distance between ground truth trajectories and generated trajectories.
- Final Displacement Error (FDE): $l_2$ distance between ground truth trajectories and generated trajectories at the end of time horizon $T$.

Since the models generate multimodal output, we report the minimum ADE and FDE from 20 randomly generated samples instead. In addition to these two commonly used metrics, we introduced two more metrics that are relatively new to the domain of human trajectory forecasting:

- Multimodality [64]: Mean $l_2$ distance between $N$ pairs of generated trajectories under same input condition. We set $N = 20$.
- Goal Rate ($M$): Proportions of generated trajectories which reach at least $M$ number of goals for the entire sequence. We let $M = 1$. (Only applicable to SSN dataset.)

Note that we perform all evaluations on low-level representations. We always decode the generated discrete actions into continuous human trajectories and run evaluations.

## 5.1. Implementation Details

For our action quantization, the codebook size is set to 256 × 128, where the number of discrete actions is 256, and the dimension of each action token is 128. We set $\beta = 1$ and $T_{vq} = 5$ for training. We use multilayer perceptrons (MLPs) as our HAQ encoders and decoder. The VQ-VAE network is optimized using AdamW [26] with a learning rate of 1e-6 and batch size of 128. For both SSN and JRDB datasets, visual images are extracted into 2D body pose features with dimensions of 50. Our discrete diffusion model has two variants: diffusion-MLP and diffusion-Transformer [39]. The diffusion-MLP is an MLP model with 3 hidden layers. And the diffusion-Transformer is a standard transformer [55] model with 2 encoder blocks and multi-head at-

| Dataset | SSN | | | JRDB | |
|---|---|---|---|---|---|
| Methods | ADE/FDE ↓ | MModality ↑ | Goal Rate ↑ | ADE/FDE ↓ | MModality ↑ |
| LSTM[67] | 1.19/2.01 | - | 0.39 | 3.71/4.78 | - |
| LSTM-CNN [67] | 1.03/1.97 | - | 0.45 | 3.51/4.61 | - |
| CVAE [30] | 0.79/1.14 | 0.19 | 0.67 | 2.12/2.77 | 0.41 |
| Transformer [62] | 0.76/1.32 | **0.27** | 0.71 | 1.89/2.85 | **0.49** |
| Diffusion-BC [39] | 0.84/1.21 | 0.24 | 0.59 | 2.33/3.13 | 0.38 |
| Transformer-discrete | 0.73/1.16 | 0.21 | 0.76 | 1.91/2.81 | 0.33 |
| Ours (MLP) | 0.72/1.19 | 0.23 | 0.75 | 1.77/2.93 | 0.36 |
| Ours (no guidance) | 0.73/1.05 | **0.27** | 0.87 | 1.63/2.61 | 0.43 |
| Ours | **0.68/0.97** | 0.25 | **0.88** | **1.49/2.53** | 0.39 |

Table 1. **Quantitative comparison** of our method and baselines with T = 30. Our model achieves the best or second-best performances on all datasets. The first-best is highlighted by **bold**, and the second-best is highlighted by underline.

| | SSN | JRDB |
|---|---|---|
| Method | ADE/FDE ↓ | ADE/FDE ↓ |
| Transformer + VQVAE | 0.79/1.39 | 1.93/2.83 |
| Ours + VQVAE | 0.77/1.20 | 1.72/2.74 |
| Diffusion-BC | 0.84/1.21 | 2.33/3.13 |
| Diffusion-BC + guidance | 0.75/1.17 | 2.01/2.89 |
| Ours (Full) | **0.68/0.97** | **1.49/2.53** |

Table 2. **Ablation studies** of quantization choice and reachability guidance on SSN and JRDB datasets.

| | SSN | |
|---|---|---|
| # of BRS | ADE/FDE ↓ | MModality ↑ |
| 1 (v = 1.5) | 0.73/1.03 | **0.27** |
| 2 (v = 1, 1.5) | 0.69/1.01 | 0.25 |
| 3 (v = 0.5, 1, 1.5) | **0.68/0.97** | 0.25 |

Table 3. **Ablation study** of different number of reachable sets on SSN dataset.

tentions. We use 10 diffusion steps and square cosine noise scheduler [36]. The diffusion network is optimized with AdamW [26] with a learning rate of 1e-4. We calculate a set of backward reachable sets (BRS) based on different maximum traveling speed assumptions. We assume a maximum turn rate of 1 rad/s, and a maximum acceleration of 0.5 $m/s^2$. The maximum speed is set to be $\{0.5, 1, 1.5\}$ m/s for SSN dataset, and $\{1, 2, 3\}$ m/s for JRDB dataset. Reachable set calculation is done with the helperOC Matlab toolbox [4]. Each learned discrete action represents 5 timesteps (1.5s). For evaluation, the observation length is set as 10 timesteps (3s) and we are predicting 6 future discrete actions, which

---

[4]https://github.com/HJReachability/helperOC

is 30 timesteps (9s).

## 5.2. Results

We seek to answer the following questions in our evaluation.

*How does our model compare to prior work in long-term human trajectory forecasting?* In Table 1, we compare our method to a large number of commonly used baselines. Our methods achieve the **best** ADE and FDE while maintaining a good level of diversity on both datasets. To further investigate our method's ability in long-term forecasting, we evaluate it across different forecasting horizons. In Figure 4, we evaluate from $T = 10$ (short-term) to $T = 30$ (long-term) and observe that as the forecasting horizon increases, our model significantly outperforms all other baselines. This result demonstrates that our model has superior performance in long-term human trajectory forecasting while still maintaining comparable performance in the short-term.

*How does diffusion policy compare to the autoregressive model?* We are curious how a diffusion model would perform against the autoregressive model when the inputs are discrete actions. Transformer-discrete is a variant of Transformer [62] which is trained with the same quantized actions as our method. As can be seen in Table 1, the variant Transformer-discrete outperforms the original model Transformer, suggesting the advantage of using discrete actions. Both our model and MLP-variant outperform the Transformer-discrete with lower ADE/FDE and higher Multimodality score, which indicates that the use of the discrete diffusion model is more effective in modeling a multimodal action distribution in discrete space.

*How effective is the reachability guidance to the performance of our model?* To understand how reachability guidance influences the performance of our model, we implement a variant of our model which does not use reachability guidance. In Table 1, both ADE and FDE increase

with the reachability guidance, but the multimodality score decreases. This suggests that there is a trade-off between physical feasibility and multimodality. We hypothesize that more diverse multimodal actions could lead to some states that are not physically feasible; our reachability guidance places a constraint on those states.

## 5.3. Ablation Study

We further perform an extensive ablation study to understand the contributions of individual components to our model performance. In Table 2, we investigate (1) the effect of our proposed hierarchical action quantization and (2) the performance of reachability guidance with the continuous diffusion model. First, we replace the hierarchical action quantization component with a simple VQ-VAE on Transformer-discrete and our method. We could see a significant decrease in performance on both datasets, indicating the importance of our hierarchical action quantization scheme. Then we integrate our reachability guidance with the continuous diffusion baseline: Diffusion-BC, resulting in a 12% performance increase. This highlights the flexibility and importance of the reachability guidance. Finally, our second ablation study Table 3 investigates the effect of the backward reachable set. As one would expect, calculating a more restrictive condition for the reachable sets can result in a slight improvement in terms of ADE and FDE with a little sacrifice in diversity. When there is only one BRS calculated with a very relaxed assumption, the model has a similar performance as the one without reachability guidance. Thus, a more carefully designed set of dynamic assumptions could maximize the performance of the reachability guidance.

## 6. CONCLUSIONS

In this work, we present our discrete diffusion framework that generates future human behaviors with physics-inspired guidance. Our goal is for robots to be able to imitate realistic and diverse human behaviors in the long term. To achieve this, we propose reachability guidance to enforce physical constraints during the diffusion process in discrete action space. The proposed reachability guidance can be used on any diffusion model without retraining. Experimental results on human trajectory forecasting datasets demonstrate the superior performance of our framework. The limitations of our framework include: the dynamics of humans, and assumptions for reachable set calculation. Recent developments in the diffusion models [42, 63] have shown promising results incorporating physics in the sampling process, we hope the proposed framework and reachability guidance could open up new directions for future work. We believe our framework is robust to other tasks, *e.g.*, autonomous driving, motion generation, etc, which are yet to be explored.

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
