# OpenReview forum: "Predicting Long-Term Human Behaviors in Discrete Representations via Physics-Guided Diffusion"
_thecvf.com/CVPR/2024/Workshop/POETS — CVPR 2024 Workshop POETS Oral_

### Official Review · Reviewer_7XMf · 2024-05-06
**Interesting Paper**

**Rating:** 7
**Confidence:** 3

**Review:**

This paper presents an interesting framework for long-term human motion prediction. The idea of combining VQ-VAE and diffusion policy makes sense. The reachability guidance is also suitable for this problem.

Suggestions: texts in the figs are not very clear. It would also be interesting to explore other conditions, e.g. social and scene.

---

### Official Review · Reviewer_As4X · 2024-05-08

**Rating:** 7
**Confidence:** 4

**Review:**

The paper is well organized, and the topic is suitable for this workshop. Therefore, I prefer to accept this paper.

---

### Official Review · Reviewer_njfc · 2024-05-13
**Review of Submission 2**

**Rating:** 8
**Confidence:** 5

**Review:**

Strengths:
1. Innovative Approach: The integration of physics-guided diffusion models with discrete action representation provides a novel approach to predicting long-term human behaviors. This method enhances the realism and feasibility of generated trajectories by incorporating physical constraints.
2. Robust Framework: The paper presents a comprehensive framework that combines Hierarchical Action Quantization (HAQ), Action Diffusion Policy, and Reachability Guidance. This multi-component strategy allows for a nuanced modeling of human behaviors that are both diverse and physically plausible.
3. Extensive Evaluation: The authors provide a detailed evaluation of two publicly available datasets, demonstrating superior performance in long-term trajectory prediction compared to several baselines. The use of both common and novel metrics enriches the validation process.
4. Theoretical and Practical Contributions: The paper makes significant theoretical contributions with its method for integrating discrete action spaces and diffusion models. Practically, it offers a new tool for improving human-robot interaction in complex scenarios.


Weaknesses:
1. Dependency on Accurate Physical Modeling: The performance of the model heavily relies on the accuracy of the physical modeling of human dynamics. Misestimations in this aspect could lead to less reliable trajectory predictions.
2. Generalization Across Different Contexts: While the model performs well on the tested datasets, its ability to generalize across different environments or more dynamic scenarios isn't fully explored. The model might perform differently in settings not covered by the datasets used.

---

### Meta-Review · Program_Chairs · 2024-05-14

**Recommendation:** Accept (Oral)
**Confidence:** 5

**Metareview:**

The paper presents a novel, physics-guided diffusion model integrated with action representation for predicting human behaviors, demonstrating significant enhancements in realism and feasibility. Strengths include its robust framework, extensive evaluation, and both theoretical and practical contributions. Weaknesses focus on its heavy reliance on accurate physical modeling and limited generalization across different contexts. Overall, it receives positive recommendations for acceptance.

---

### Decision · Program_Chairs · 2024-05-14

Accept (Oral)